# Targeting Extracellular RNA Mitigates Hepatic Lipotoxicity and Liver Injury in NASH

**DOI:** 10.3390/cells12141845

**Published:** 2023-07-13

**Authors:** Archana Tewari, Sangam Rajak, Sana Raza, Pratima Gupta, Bandana Chakravarti, Jyotika Srivastava, Chandra P. Chaturvedi, Rohit A. Sinha

**Affiliations:** 1Department of Endocrinology, Sanjay Gandhi Postgraduate Institute of Medical Sciences, Lucknow 226014, India; archanatewari94@gmail.com (A.T.); rajaksangam@gmail.com (S.R.); raza.sana9@gmail.com (S.R.); pratimagupta2009@gmail.com (P.G.); vandanaks@gmail.com (B.C.); 2Stem Cell Research Facility, Department of Hematology, Sanjay Gandhi Postgraduate Institute of Medical Sciences, Lucknow 226014, India; jyotikasri89@gmail.com (J.S.); chaturvedicp75@rediffmail.com (C.P.C.)

**Keywords:** damage-associated molecular patterns (DAMPs), extracellular RNA, lipotoxicity, inflammation, non-alcoholic steatohepatitis (NASH)

## Abstract

Non-alcoholic steatohepatitis (NASH) is a clinically serious stage of non-alcoholic fatty liver disease (NAFLD). Histologically characterized by hepatocyte ballooning, immune cell infiltration, and fibrosis, NASH, at a molecular level, involves lipid-induced hepatocyte death and cytokine production. Currently, there are very few diagnostic biomarkers available to screen for NASH, and no pharmacological intervention is available for its treatment. In this study, we show that hepatocyte damage induced by lipotoxicity results in the release of extracellular RNAs (eRNAs), which serve as damage-associated molecular patterns (DAMPs) that stimulate the expression of pro-apoptotic and pro-inflammatory cytokines, aggravate inflammation, and lead to cell death in HepG2 cells. Furthermore, the inhibition of eRNA activity by RNase 1 significantly increases cellular viability and reduces NF-kB-mediated cytokine production. Similarly, RNase 1 administration significantly improves hepatic steatosis, inflammatory and injury markers in a murine NASH model. Therefore, this study, for the first time, underscores the therapeutic potential of inhibiting eRNA action as a novel strategy for NASH treatment.

## 1. Introduction

Non-alcoholic fatty liver disease (NAFLD) refers to a spectrum of liver conditions affecting people who drink little to no alcohol [1]. This ranges from benign hepatic fat deposition (steatosis) to a much more severe condition known as non-alcoholic steatohepatitis (NASH) [2]. NASH is a clinically alarming stage of NAFLD that results in liver injury, fibrosis, and cirrhosis and may eventually culminates into hepatocellular cancer (HCC) [2]. Although various drugs are in the pipeline [3], currently, we have no approved drug therapy for NASH, and its progression may become irreversible in some individuals.

Hepatic injury due to lipotoxicity associated with NASH is triggered by lipids, which include saturated free fatty acids such as palmitic acid (PA) [4]. Upon entry into hepatocytes, PA induces a number of cellular derangements, including activation of stress kinases, oxidative stress, mitochondrial dysfunction [5], and, eventually, cell lysis [6,7]. The death of hepatocytes caused by lipotoxicity results in the release of a variety of damage-associated molecular patterns (DAMPs), such as DNA fragments, histones, ATP, uric acid, and cholesterol crystals, which act as stress signals to activate DAMP receptors, such as Toll-like receptors (TLRs), purinoreceptors (P2X and P2Y), and C-type lectin domain (CLEC)12A, thereby initiating sterile inflammation and exacerbating tissue injury in NASH [8,9,10,11,12,13]. Extracellular RNAs (eRNAs) also belong to this heterogeneous group of DAMPs and include several types of RNAs present in the extracellular environment, including microRNA (miRNA), transfer RNA (tRNA), small interfering RNA (siRNA), and long noncoding RNA (lncRNA) [14]. eRNA released because of tissue injury may play a role in differentiation, chromatin modification, and inflammation, as well as tissue injury and repair [14,15,16,17,18,19,20,21,22]. eRNA binds to the Toll-like receptor (TLR) family members to induce apoptosis and inflammation in diseases such as ischemia-reperfusion injury and sepsis injury in cardiomyocytes [23,24,25]. Consistently, the administration of Ribonuclease 1 (RNase 1), which degrades eRNA, has shown efficacy in decreasing disease severity in several preclinical disease models [23,24,25]. However, it is unknown whether eRNA plays a role in hepatic cell injury and pro-inflammatory cytokine production in NASH and whether RNase 1 administration can act as an interventional strategy to reduce NASH progression.

In this study, we demonstrate that eRNA released from injured/dead hepatocytes upon lipotoxic insult increases the severity of hepatic injury, along with increased pro-inflammatory cytokine production. Furthermore, the degradation of eRNA by RNase 1 treatment attenuates lipotoxicity both in vitro and in a mouse model of NASH.

## 2. Materials and Methods

### 2.1. Cell Culture and Treatment

HepG2 and AML12 hepatic cells were procured from the American Type Culture Collection (ATCC, Manassas, VA, USA). HepG2 cells were cultured in DMEM supplemented with 10% fetal bovine serum (FBS) and 1% penicillin/streptomycin. Mouse hepatocytes, AML12 cells (CRL-2254), were maintained in DMEM-F12 1:1 containing 10% FBS, supplemented with ITS (Thermo Fisher Scientific, Waltham, MA, USA, 41400), 10 nM of dexamethasone, and penicillin/streptomycin. Both cell lines were maintained at 37 °C in the presence of 5% CO_2_. At 80% confluence, lipotoxicity was induced in the cells using 0.75 mM of PA (saturated fatty acid), dissolved in ethanol; 0.5% BSA was used as a carrier for PA. Cells in the vehicle group were given only ethanol in a 0.5% BSA-containing medium. For the administration of RNase 1, cells were incubated with 2.8 U/mL of RNase 1 one hour prior to the addition of PA, and the treatment was continued up to 24 h to maintain a sustained release of RNase 1 (Invitrogen, Waltham, MA, USA, #12091021) in the culture medium.

### 2.2. MTT Assay

HepG2 and AML12 cells (1 × 10^4^), seeded into 96-well plates, were treated with PA with and without RNase 1. Vehicle control cells were given only ethanol in a 0.5% BSA-containing medium. After 24 h of treatment, the medium was supplemented with MTT reagent (5 mg/mL) for 4 h at 37 °C. DMSO (100 uL) was then added to dissolve the purple formazan crystals, and the plate was shaken for 10 min. Absorbance was recorded at 570 nm using a multi-well spectrophotometer (ELISA reader).

MTT assay was also performed in HepG2 cells treated with PA in the presence and absence of 10 uM of TLR3/dsRNA complex inhibitor (Merck, Darmstadt, Germany, #614310). In this case, cells in the vehicle group received both DMSO and ethanol in the 0.5% BSA-containing medium as the TLR3/dsRNA complex inhibitor was dissolved in DMSO.

### 2.3. Analysis of Total eRNA Levels

Total eRNA levels in the conditioned media from vehicle control, PA-treated, and PA + RNase 1-treated HepG2 cells were analyzed using an eRNA quantitation kit (Promega, Cat No: E3310 QuantiFluor^®^ RNA System, Madison, WI, USA), according to the manufacturer’s protocol, using a multimode reader.

### 2.4. Oil Red O Staining

Lipid droplets in HepG2 cells were assayed using Oil Red O stain. The 96-well plates containing HepG2 cells were washed twice with phosphate-buffered saline. The cells were fixed with 10% paraformaldehyde for 30 min. After this, the cells were washed with distilled water and 60% isopropanol was added for 5 min. The cells were then incubated in Oil Red O stain for 20 min, followed by washing with distilled water to remove excess stain. Next, the cells were incubated with Hematoxylin for 1 min. Oil Red O stain was extracted with 100% isopropanol for 5 min, and absorbance was recorded at 492 nm using a multi-well spectrophotometer (ELISA reader).

### 2.5. Animal Experiments

HFMCD NASH model: 6–8-week-old male C57BL/6N mice were fed with a high-fat methionine-choline-deficient (HFMCD) diet consisting of 60 kcal% fat and 0.1% methionine by weight (A06071302; Research Diets, New Brunswick, NJ, USA). Littermate mice fed with normal chow diet (NCD) were used as the control mice. The mice were divided into three groups: NCD, HFMCD diet alone for 4 weeks, and HFMCD diet with RNase 1 (50 ug/kg). Rnase1 was injected intra-peritoneally, starting every alternate day after 2 weeks of feeding on the HFMCD diet, and continued for the next 2 weeks. All animals were sacrificed after 4 weeks, and serum and liver tissues were collected. All animal procedures were carried out in accordance with the institutional guidelines for animal research at SGPGIMS.

### 2.6. RNA Isolation and qRT-PCR

Total RNA from the mouse liver tissues and HepG2 cells was isolated using Trizol reagent (#343909). qRT-PCR was performed using the QuantiTect SYBR Green PCR Kit (Qiagen, 204141) according to the manufacturer’s instructions. The GAPDH gene was used for normalization. KiCqStart SYBR Green Primers were purchased from Sigma-Aldrich, St. Louis, MO, USA. The primer IDs were Mouse Tnf-alpha (M_Tnf_1), Mouse IL6 (M_IL6_1), Mouse IL1 beta (M_IL 1 beta _1), Mouse CCL20 (M_CCL20_1), Mouse Ccl2 (M_Ccl2_1), Mouse Ccl3 (M_Ccl3_1), Mouse Cxcl10 (M_Cxcl10_1)*,* Mouse GAPDH (M_ Gapdh_1), Human Tnf-alpha (H_Tnf_1), Human IL-6 (H_IL6_1), Human CCL3 (H_CCL3L3_1), Human CXCL10 (H_CXCL10_1), Human CCL20 (H_CCL20_1), and Human GAPDH (H_Gapdh_1).

### 2.7. Western Blotting

The HepG2 cells and mouse liver tissue samples were lysed using RIPA buffer (Sigma, Kanagawa, Japan, #R0278) mixed with phosphatase and protease inhibitor cocktails. Fifty microgram of protein was subjected to sodium dodecyl sulfate–polyacrylamide gel electrophoresis and electro-transferred onto a nitrocellulose membrane as per the manufacturer’s guidelines (Bio-Rad Laboratories, Hercules, CA, USA). Image acquisition was performed using ChemiDoc (Syngene Model No: -G: BOX CHEMIXRQ). Densitometry analysis was performed using the ImageJ software (NIH, Bethesda, MD, USA). The antibodies used were anti-GAPDH (Cell Signaling Technology, Danvers, MA, USA, # 5174), anti-F4/80 (Cell Signaling Technology, # 70076), anti-Cleaved PARP (Cell Signaling Technology, #5625), anti-Phospho-c-JUN (Cell Signaling Technology, #3270), anti-Phospho-p-38MAPK (Cell Signaling Technology, #4511), anti-Phospho-SAPK/JNK (Cell Signaling Technology, #4668), anti-JNK (Cell Signaling Technology, #9252), anti-c-JUN (Cell Signaling Technology, #9165), anti-p-38MAPK (Cell Signaling Technology, #8690), anti-LC3B (Cell Signaling Technology, #2775), and anti-P62 (Cell Signaling Technology, #39749).

### 2.8. H&E Staining

Tissues from the liver were fixed in situ after opening the abdomen, immersed in 4% paraformaldehyde in PBS, and processed into paraffin blocks. The sections were sliced at 5-micron thickness and mounted on slides to perform H&E staining. The images were analyzed using the ImageJ software, NIH. The non-alcoholic fatty liver disease activity scoring (NAS) system was used to quantify steatosis (0–3), lobular inflammation (0–3), and hepatocellular ballooning (0–2). Liver biopsies scoring an NAS ≥5 were classified as definitive NASH [26].

### 2.9. Serum ALT Test

Alanine aminotransferase (ALT) levels in the serum were assessed using a kit (Abcam calorimetric assay kit #ab105134) as per the manufacturer’s protocol.

### 2.10. Liver TG Analysis

Liver triglyceride (TG) levels were assessed using a kit (Cayman, Triglyceride Colorimetric assay kit # Cat No. 10010303) as per the manufacturer’s protocol.

### 2.11. Confocal Microscopy

Immunofluorescence experiments were performed on treated HepG2 cells in chambered slides and in paraffin-embedded sections of mouse liver. In brief, formalin-fixed cells and tissues were permeabilized with 0.1% Triton X-100 (Sigma-Aldrich, X100) in PBS for 5–10 min and blocked with 3% BSA-PBS for 30 min, at room temperature. The cells and tissues were incubated with the primary antibody (1:200 in 3% BSA-PBS) overnight at 4 °C, followed by fluorochrome-labeled secondary antibodies (Molecular probes), and cell imaging was performed using an LSM710 Carl Zeiss (Carl Zeiss Microscopy GmbH, Munich, Germany) confocal microscope. The primary antibody used for HepG2 cells and liver sections was anti-NFkB (Cell Signaling Technology, #8242). Colocalization studies were performed using the ImageJ software with JACoP plugin.

### 2.12. Mitochondrial ROS Detection

Mitochondrial ROS was detected in vehicle control and treated HepG2 cells using 5 µM of MitoSOX™ Red (catalog number #M36008), in which the cells were incubated for 20–30 min at 37 °C. After washing with PBS, reading was taken using a fluorimeter at an excitation wavelength of 510 nm and an emission wavelength of 580 nm.

### 2.13. Mitochondrial Membrane Potential (MMP)

Mitochondrial membrane potential (MMP) was assessed via JC-1 staining. HepG2 cells seeded into 96-well black bottom plates were treated with PA with and without RNase 1 for 24 h. The cells were washed with PBS and incubated with 2 µM of JC-1 dye (Invitrogen, catalog number #T3168) at 37 °C for 30 min. Fluorescence was measured using a fluorimeter (Synergy HTX multimode microplate reader, BioTek) at an excitation wavelength of 488 nm and emission wavelengths of 530 nm (monomer) and 590 nm (aggregates). Loss of MMP was calculated from the ratio of emission wavelength at 590 nm/530 nm.

### 2.14. Statistics

One-way ANOVA with Tukey’s post hoc test was used to compare among groups. Statistical analysis was performed using the Graph-Pad Prism software, version 5.0. * *p* < 0.05, ** *p* < 0.01, and *** *p*< 0.001 were considered statistically significant.

## 3. Results

### 3.1. RNase 1 Attenuates PA-Induced Cellular Injury in HepG2 Cells

Lipotoxicity caused by PA is known to induce apoptosis and cellular stress in hepatic cells, and its effect mimics hepatocyte damage in NASH [27]. To determine if PA-induced hepatocyte injury leads to the release of eRNA, we assessed the eRNA content in the conditioned media of PA-treated HepG2 cells and observed a significant increase in the eRNA levels. Furthermore, the co-administration of RNase 1 significantly reduced the released eRNA levels upon PA treatment, when compared with PA treatment alone (Figure 1A). Concurrently, the RNase 1 treatment also significantly rescued the cellular viability of HepG2 cells upon lipotoxic insult (Figure 1B). These results were further supported by the decrease in cleaved PARP levels (an apoptosis marker) in RNase 1-and-PA-treated cells, when compared to the increased apoptosis observed in PA-treated cells (Figure 1C,D). Additionally, PA-induced hepatic steatosis was also reduced by RNase 1 administration (Appendix A). Cellular stress kinases, such as p-JNK and p-p38MAPK, are key mediators of PA-induced cellular stress [28]. In line with these observations, we also assessed the effect of the RNase 1 treatment on PA-induced JNK and p38MAPK activation. Our results showed that, indeed, similar to its effect on cellular viability, the targeted degradation of eRNA by RNase 1 significantly reduced the PA-induced activation of JNK, its downstream target c-JUN, and p38MAPK (Figure 1E–K). Notably, although RNase 1 restored p-JNK and p-p38MAPK back to the control levels, its effect did not restore p-c-JUN back to the control levels even though p-c-JUN levels were significantly rescued in RNase 1 and PA-treated cells. Furthermore, the RNase 1 treatment also rescued PA-induced autophagy block, which is a major mediator of lipotoxicity [29] (Appendix A). To verify if the observed effects of RNase 1 are specifically due to its effects on eRNA, we also used a pharmacological inhibitor of TLR3, which is a receptor of eRNA, and observed a similar rescue effect on PA-induced cell death (Figure 1L). This small-molecule inhibitor specifically inhibits the formation of TLR3/dsRNA complex and has been shown to prevent eRNA signaling [30]. Concurrently, the administration of RNase 1 alone to HepG2 cells had no effect on cell viability (Appendix A). These sets of results, thus, rule out any possible off-target effect of RNase 1. We further validated the effect of RNase 1 on another hepatocyte cell line (AML-12) and observed effects similar to those in HepG2 cells (Appendix A).

PA is also known to increase mitochondrial dysfunction via its effect on mitochondrial reactive oxygen species (ROS) generation and loss of MMP [5]. Therefore, to investigate the effect of RNase 1 on PA-induced mitochondrial damage, we first assessed the levels of mitochondrial ROS and observed significant protection conferred by RNase 1 to PA-induced mitochondrial ROS generation (Figure 2A). Similarly, in agreement with its effect on mitochondrial ROS, RNase 1 prevented PA-induced reduction in MMP in HepG2 cells, as measured using JC-1 staining (Figure 2B).

Therefore, collectively, these data indicate a direct involvement of eRNA in PA-induced hepatic injury in vitro and the abrogation of these effects by RNase 1 treatment.

### 3.2. RNase 1 Attenuates PA-Induced Pro-Inflammatory Signaling in HepG2 Cells

PA-induced proinflammatory and pro-apoptotic signaling is linked to hepatic inflammation and cell death in NASH [31], and it is also associated with NF-κB activation [32]. Intriguingly, eRNA released from injured cells have been previously shown to induce TLR receptors and stimulate NF-κB-dependent cytokine and chemokine synthesis [14]. Therefore, we speculated that PA-induced cytokine production is partly mediated by eRNA signaling. To test this, we treated HepG2 cells with PA alone or PA along with RNase 1 and assessed the activation of NF-κB along with pro-inflammatory cytokine expression. Our results definitively showed that the RNase 1 treatment almost completely inhibited PA-induced NF-κB nuclear translocation, which measures its activation (Figure 3A,B). In concurrence with NF-κB nuclear translocation, RNase 1 also prevented PA-induced activation of NF-κB transcriptional targets *IL-6* and *TNFα* in the presence of PA (Figure 3C). As *TNFα* is a major cell death inducer in hepatic cells [31], its inhibition may also explain the observed increase in the viability upon RNase 1 administration as described earlier (Figure 1B). Similarly, the PA-induced expression of several other NF-κB-induced chemokines, such as *CCL3*, *CCL20*, and *CXCL10*, were repressed by the RNase 1 co-treatment (Figure 3D). Therefore, these data suggest an anti-inflammatory action of RNase 1 in hepatic cells treated with PA.

### 3.3. RNase 1 Administration Mitigates NASH-Induced Liver Injury in Mouse

To determine if eRNA is also involved in NASH pathogenesis in vivo, we used C57BL/6N mice and fed them the HFMCD diet supplemented with 60 kcal% fat and 0.1% methionine by weight. These animals rapidly developed severe NASH phenotype, owing to the lipotoxic action of stored hepatic fat, when compared to the mice fed with the control diet [33,34,35]. We administered RNase 1 to the animals fed with the HFMCD diet and followed its effects on the NASH phenotype. RNase 1 (50 ug/kg) was injected i.p. every alternate day starting after two weeks of HFMCD feeding and continued for the next two weeks. The mice fed either the NCD or HFMCD diet for four weeks were used as comparison. All the animals were sacrificed after four weeks.

Histological evaluation of the livers of the mice fed with the NC, HFMCD diet, and HFMCD diet with RNase 1 intervention showed a marked increase in hepatic steatosis in the HFMCD diet-fed group, and its reduction upon RNase 1 co-administration (Figure 4A). These results were further corroborated by hepatic TG measurement in the different experimental groups (Figure 4B). Consumption of the HFMCD diet damages hepatic tissue, which is reflected by the elevated levels of serum ALT in the HFMCD diet-fed mice, a result that is consistent with the histological findings (Figure 4C). However, RNase 1 administration significantly reduced the levels of ALT in the mice fed with the HFMCD diet (Figure 4C). Additionally, cellular stress markers such as p-JNK and p-p38MAPK, which were upregulated in the livers of the HFMCD diet-fed mice, were also reduced by RNase 1 administration (Appendix A). Therefore, these results indicate a protective effect of RNase 1 on lipid-induced liver injury in vivo.

### 3.4. RNase 1 Administration Reduces NASH-Induced Liver Inflammation

The HFMCD mouse model mimics several aspects of human NASH pertaining to inflammation and immune cell infiltration [36]. Therefore, to validate our in vitro findings, we looked at the effect of RNase 1 on NASH-induced NF-κB activation and pro-inflammatory cytokine expression. Our results demonstrated that similar to the effect of RNase 1 observed in HepG2 cells, in vivo blocking of eRNA action by RNase 1 significantly prevented hepatic NF-κB nuclear translocation (Figure 5A,B) and the expression of pro-inflammatory genes (Figure 5C,D and Appendix A) in the liver of the HFMCD diet-fed mice. Additionally, the mRNA expression of chemokines was also similarly altered by RNase 1 administration in the liver of HFMCD diet-fed mice (Figure 5E and Appendix A). Furthermore, we observed an inhibitory effect of RNase 1 administration on intrahepatic macrophage infiltration in the liver of the HFMCD diet-fed mice as assessed by hepatic F4/80 levels (Figure 5F,G). The outcomes regarding steatosis, inflammation, and hepatic ballooning were assessed histologically using the NAFLD activity score, which significantly decreased upon RNase 1 treatment (Figure 5H). Collectively, these results demonstrate the anti-lipotoxic action of RNase 1 via attenuating NASH-induced tissue damage and inflammation.

## 4. Discussion

As a disease, NAFLD displays a spectrum of disease stages ranging from benign steatosis to NASH [2], which is further presented as steatohepatitis, fibrosis, and cirrhosis [1]. Unfortunately, despite its increasing prevalence across the globe, we still do not have an effective treatment strategy for NASH other than life-style modifications [37]. At the molecular level, a tipping point during progression from benign steatosis to NASH is the increase in the incidence of hepatocyte injury, which is associated with necrotic/apoptotic death of hepatocytes, release of pro-inflammatory cytokines, and infiltration of macrophages leading to inflammation [38]. Therefore, understanding these events that link hepatocyte injury to inflammation is the key to preventing the transition and progression of steatosis to NASH.

One of the key concepts that helps to explain the large spectrum of NAFLD stages seen in patients is the process called “Lipotoxicity” [38]. Indeed, when a liver cell experiences an increased flux of lipids, it attempts to store these into inert lipid droplets; however, in the case when the lipid load overruns the adaptive capacity of hepatocytes, fatty acids inside these cells lead to the generation of reactive lipid species, which triggers cell death via several distinct molecular mechanisms, encompassing the process of lipotoxicity [39]. Lipotoxic injury results in the release of DAMPs by hepatocytes, also known as danger signals that trigger the activation of sterile (i.e., in the absence of infection) inflammatory pathways, which, when perpetuated over time, result in chronic injury and an abnormal wound healing response with fibrosis [40,41].

Basic and clinical research studies have provided evidence that eRNA released by injured cells is an important player in the crosstalk between immunity and tissue injury in several diseases [14]. In fact, (patho-)physiological functions of eRNA are associated with, and in many cases, causally related to conditions such as arterial and venous thrombosis, atherosclerosis, ischemia/reperfusion injury, and tumor progression, especially with the elevated inflammatory status of these diseases [17,20,42,43,44,45]. eRNAs can be liberated from cells in a free form or bound to proteins or phospholipids, as well as in association with extracellular vesicles (EVs) or apoptotic bodies [14]. Once released from an injured cell, eRNAs bind to adjacent cells via surface-bound Toll-like receptors (TLRs) and activate the pro-inflammatory NF-kB pathway, leading to the synthesis and secretion of pro-inflammatory cytokines like TNF-α and IL-6 [14]. Therefore, eRNA may serve as a catalyst to increase the inflammatory environment and worsen disease pathology. However, the adverse effects of eRNA are countered by several extracellular RNases, which proteolytically degrade eRNA [14]. In this regard, the exogenously administered RNase 1 bears considerable potential as new therapeutic agent based on its tissue-protective functions that may translate into anti-inflammatory properties in different pathological situations [14].

As NASH represents a milieu wherein injured hepatocytes and cytokine production co-exist, it may be possible that eRNA drives NASH progression; however, this possibility still has not been tested so far. In the results presented here, we provide, for the first time, direct evidence of eRNA acting as a paracrine pro-inflammatory mediator of lipotoxicity in hepatic cells. Our results demonstrate that inhibiting the biological activity of eRNA by RNase 1 prevents PA-induced cellular stress in HepG2 cells as well as the expression of pro-inflammatory cytokines. Additionally, these results were confirmed in an animal model of NASH, which showed that RNase 1 administration significantly reduced liver injury and inflammation associated with diet-induced NASH. Our study is in agreement with a previous report showing the possible involvement of TLR3 in human NASH [46], which further strengthens our proposition of eRNA-mediated inflammation in NASH.

Further studies are needed to profile eRNA across stratified NAFLD patients to assess the differences associated with the progression of this disease. Additionally, further experiments will be required to engineer stable RNase 1 with minimum side effects, which can enter human clinical trials for NASH. In line with this, RNase therapy is already in clinical trials for cancer treatment [47]. Furthermore, based on our in vitro results, the use of specific TLR3 inhibitors should also be tested in preclinical models of NASH.

In summary, our results suggest that the release of eRNAs by injured hepatocytes under a lipotoxic environment constitutes a feed-forward loop, wherein hepatocyte injury is intertwined with the maintenance of a pro-inflammatory environment. Therefore, targeting eRNAs by RNase 1 or specific TLR3/dsRNA inhibitors may be a new therapeutic strategy to reduce the pathogenesis of NASH (Figure 6).

## 5. Conclusions

Our results provide an insight into the pathological role of eRNAs released by injured hepatocytes and put forth a proof of concept for limiting eRNA action to blunt NASH progression.

## Figures and Tables

**Figure 1 cells-12-01845-f001:**
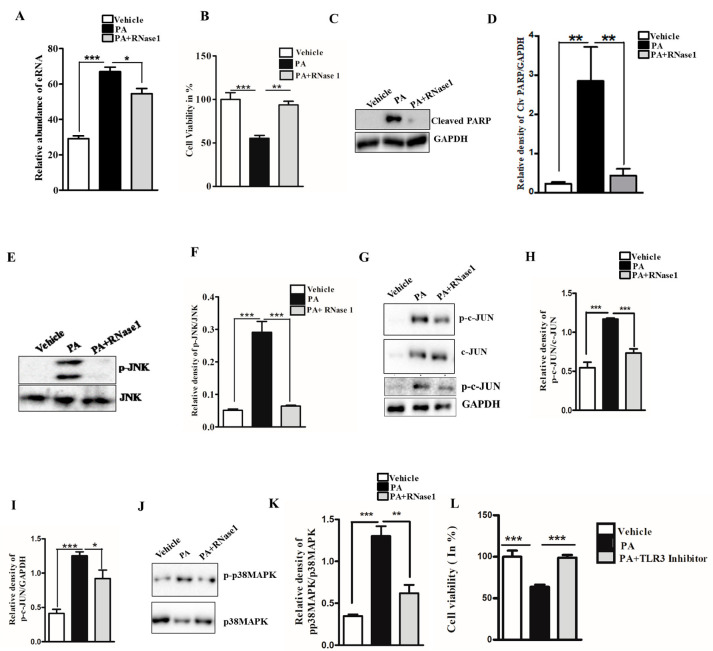
**eRNA mediates lipotoxic injury in HepG2 cells.** (**A**) Relative abundance of eRNA in different experimental groups (vehicle, PA, and PA + RNase 1). Values are expressed as mean ± SEM, n = 5; *** *p*< 0.001 when comparing vehicle to PA, and * *p* < 0.05 when comparing PA to PA+ RNase 1. (**B**) % cell viability observed using MTT assay among the three experimental groups. Data are presented as mean ± SEM, n = 5; *** *p* < 0.001 when comparing vehicle to PA, and ** *p*< 0.01 when comparing PA to PA +RNase 1. (**C**,**D**) Representative immunoblots and densitometric analysis showing protein levels of cleaved PARP in different experimental groups. Values are expressed as mean ± SEM, n = 5; ** *p*< 0.01 when comparing vehicle to PA and PA to PA+ RNase 1. (**E**–**K**) Representative immunoblots and densitometric analysis showing protein levels of p-JNK, p-c-JUN, and p-p38MAPK in different experimental groups. Values are expressed as mean ± SEM, n = 5, * *p* < 0.05 and *** *p* < 0.001, when PA is compared to vehicle and PA + RNase 1. (**L**) % cell viability observed using MTT assay among vehicle, PA-treated, and PA + TLR3 inhibitor-treated HepG2 cells. Data are presented as mean ± SEM, n = 5 ***, *p* < 0.001 when comparing vehicle to PA and PA to PA + RNase 1.

**Figure 2 cells-12-01845-f002:**
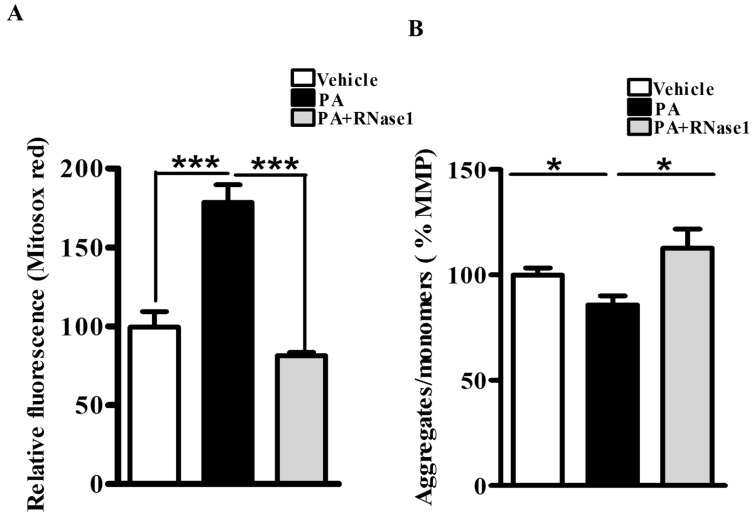
**eRNA release is associated with mitochondrial dysfunction in HepG2 cells.** (**A**) MitoSox staining showing mitochondrial ROS generation in different experimental groups (vehicle, PA, and PA + RNase 1). Values are expressed as mean ± SEM, n = 5, *** *p* < 0.001, when PA is compared to vehicle and PA +RNase 1. (**B**) JC-1 staining showing % MMP among the three experimental groups. Data are presented as mean ± SEM, n = 5, * *p* < 0.05 when comparing vehicle to PA and PA to PA + RNase 1.

**Figure 3 cells-12-01845-f003:**
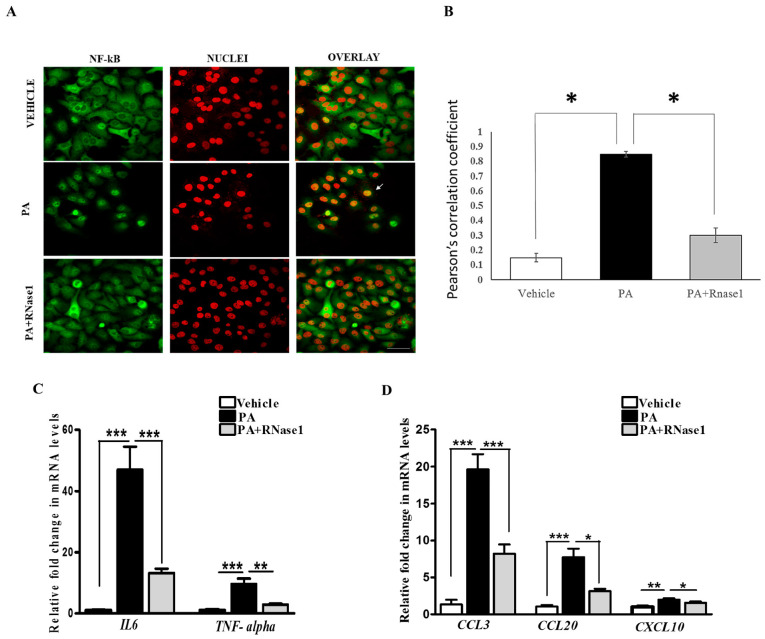
**eRNA induces the expression of pro-apoptotic and pro-inflammatory cytokines in HepG2 cells.** (**A**,**B**) Representative immunofluorescence images and Pearson’s correlation coefficient values showing co-localization of NFkB (green) and nuclei (red) in vehicle, PA-treated, and PA + RNase1-treated HepG2 cells cultured for 24 h. Values are expressed as mean ± SD (n = 5, * *p* < 0.05). Scale bar represents 10 µm. (**C**) Quantitative real-time polymerase chain reaction (qRT-PCR) analysis showing expression of pro-inflammatory cytokines (*TNF-α* and *IL6*) in VC, PA-treated, and PA + RNase1-treated HepG2 cells. Values are expressed as mean ± SEM, n = 5, *** *p* < 0.001, when PA is compared to vehicle and PA + RNase 1. (**D**) qRT-PCR analysis of pro-inflammatory chemokines (*CCL3*, *CCL20,* and *CXCL10*) in vehicle, PA-treated, and PA + RNase1-treated Hepg2 cells. Data are presented as mean ± SEM, n = 5; * *p* < 0.05, ** *p* < 0.01, and *** *p* < 0.001, when PA is compared to vehicle and PA + RNase 1.

**Figure 4 cells-12-01845-f004:**
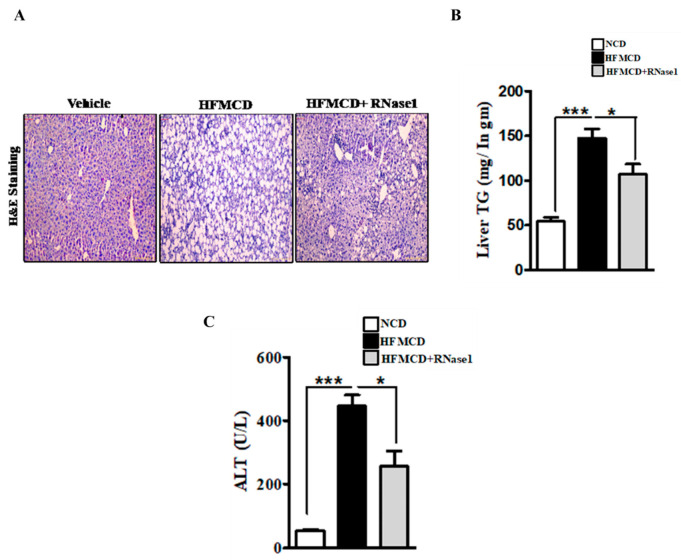
**eRNA inhibition prevents NASH-associated injury in mouse liver.** (**A**) Paraffin-fixed liver sections stained with Hematoxylin/Eosin, showing hepatic steatosis in the three experimental groups—NCD, HFMCD, and HFMCD + RNase1. (**B**) Hepatic TG levels from different experimental groups—NCD, HFMCD and HFMCD + RNase1. Values are expressed mean ± SEM, n = 5, * *p* < 0.05 and *** *p* < 0.001, when HFMCD is compared to NCD and HFMCD + RNase 1. (**C**) Serum ALT levels from the three different experimental groups. Values are expressed as mean ± SEM, n = 5, * *p* < 0.05 and *** *p* < 0.001, when HFMCD is compared to NCD and HFMCD + RNase 1.

**Figure 5 cells-12-01845-f005:**
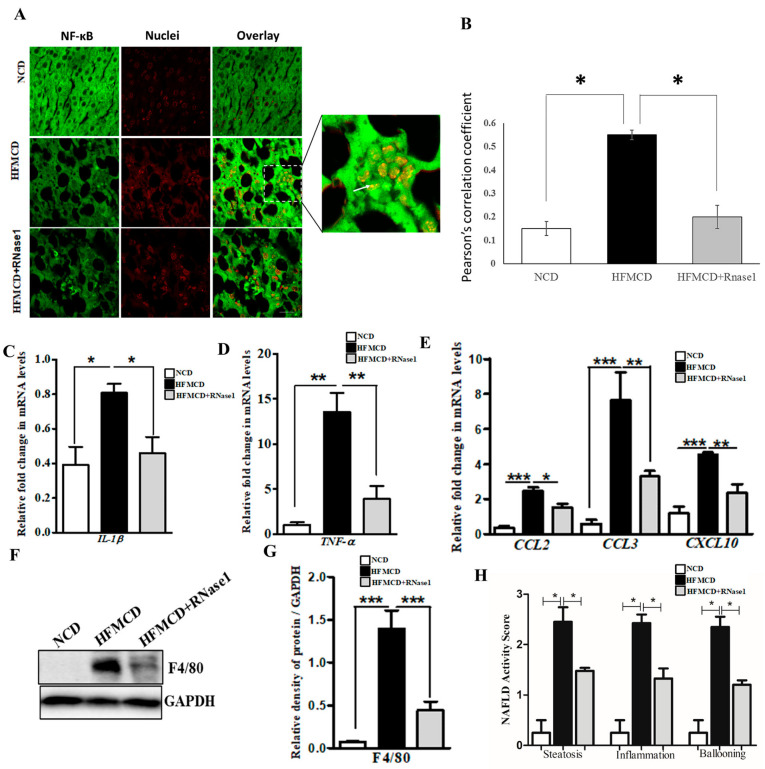
**eRNA inhibition by RNase 1 attenuates cytokine production in NASH.** (**A**,**B**) Representative immunofluorescence images and Pearson’s correlation coefficient values showing the co-localization (indicated by white arrow) of NFkB (green) and nuclei (red) in the liver sections of mice fed with the NCD, HFMCD diet, or HFMCD diet + RNase1. Values are expressed as mean ± SD (n = 5, * *p* < 0.05). Scale bar is 10 µm. (**C**,**D**) qRT-PCR analysis showing expression levels of pro-inflammatory cytokines (*IL1β* and *TNF-α*) in the livers of mice fed with the NCD, HFMCD diet, or HFMCD diet + RNase1. Values are expressed mean ± SEM, n = 5, * *p* < 0.05 and ** *p* < 0.01, when HFMCD is compared to NCD and HFMCD+RNase1. (**E**) qRT-PCR analysis showing expression levels of pro-inflammatory chemokines (*CCL2*, *CCL3*, and *CXCL10*) in the livers of mice fed with the NCD, HFMCD diet, or HFMCD diet + RNase1. Values are expressed mean ± SEM, n = 5, * *p* < 0.05, ** *p* < 0.01, and *** *p*< 0.001, when HFMCD is compared to NCD and HFMCD + RNase 1. (**F**,**G**) Representative immunoblot and densitometric analysis showing protein levels of F4/80 in different experimental groups. Values are expressed as mean ± SEM, n = 5, *** *p* < 0.001, when HFMCD is compared to NCD and HFMCD + RNase 1. (**H**) NAFLD activity score (NAS) in the three experimental groups, NCD, HFMCD and HFMCD + RNase1. Values are expressed as mean ± SEM, n = 5, * *p* < 0.05, when HFMCD is compared to NCD and HFMCD + RNase 1. In all figures, the white bar represents the NCD group, the black bar represents the HFMCD group, and light gray bar represents the HFMCD + RNase 1 group.

**Figure 6 cells-12-01845-f006:**
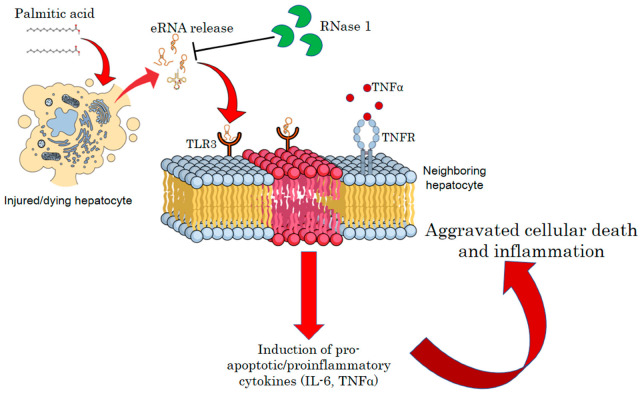
**A schematic model of eRNA-mediated lipotoxicity in hepatocytes.** Release of eRNAs from dying and injured hepatocytes upon lipotoxic insult by palmitic acid. These eRNAs bind to TLR3 receptors on the plasma/endosomal membrane of neighboring hepatocytes, inducing pro-inflammatory cytokine signaling by activating IL6 and TNF-alpha, which, in turn, aggravates cellular death and inflammation in NASH.

## Data Availability

The data presented in this study are available from the corresponding author upon request.

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
