# Peer review of "Targeting Extracellular RNA Mitigates Hepatic Lipotoxicity and Liver Injury in NASH"

_cells, 2023, doi:10.3390/cells12141845_

Round 1

Reviewer 1 Report

Comment:

In the manuscript entitled “Targeting Extracellular RNA Mitigates Hepatic Lipotoxicity” by Tiwari et al, the authors have demonstrated the role of eRNA in hepatotoxicity and subsequently, the therapeutic potential of eRNA inhibition for the treatment of NASH. The authors have used both in-vivo and in-vitro models to determine the role of eRNA-mediated hepatotoxicity. The study is well-planned and experiments are also well-performed using suitable molecular techniques. The study is relevant as there is a need to develop diagnostic markers as well as therapeutic targets for NASH.  In general, the study is well executed, the results are well presented, the manuscript is written clearly, and the conclusions are adequately supported by the data. However, I have the following comment which can be addressed to improve the study.

1.       Autophagic flux is inhibited in NASH and by lipotoxicity in cells ( Tanaka et al.; Hepatology.2016 Dec;64(6):1994-2014). Have the authors looked at autophagy in eRNA-inhibited HepG2 cells? Can RNase 1 rescue palmitate-induced autophagy block in these cells?

2. If authors can make the title more specific, it will better represent the study. The present title although suitable appears a bit generalized.

Reviewer 2 Report

In the manuscript titled "Targeting Extracellular RNA Mitigates Hepatic Lipotoxicity", the authors demonstrated that eRNA released from injured/dead hepatocytes upon lipotoxic insult, increases hepatic injury along with pro-inflammatory cytokine production. In addition, the degradation of eRNA by RNase 1 treatment attenuated lipid toxicity. This is an interesting paper that illustrates the role of eRNA in NASH. However, this manuscript is not ready for publication in its current state and needs substantial modification. there are several points that need to be addressed:

Major comments:

1) In Figure 1

a. The authors focused on the role of eRNA in lipotoxic injury of HepG2 cells. Therefore, I suggest that the authors analyze lipid profiles after PA or PA+RNase1 treatment, e.g., Oil red O staining, lipid profile-related kits (TC, TG …)

b. Figure 1G-J, the authors should analyze relative density of p-c-JUN/c-JUN and pp38MAPK/p38MAPK.

c. Figure 1 K, according to reports in the literature, the authors only analyzed the TLR3, whether the eRNA binds to other TLRs.

2) In Figure 5

a. Figure 5A, the overlay magnification of HFMCD group is different from that of other groups. In addition, the nuclei in the HFMCD group were not obvious, please explain.

b. Figure 5F, for F4/80 of the tissue, I recommend using immunohistochemical for analysis. In addition, according to the instructions of F4/80 (Cell Signaling Technology, #70076), F4/80 is broad bands in cells, while a single band is found in liver tissue in the author's results. Whether the authors choose only one band for analysis, or show a single band due to the differences of different tissues, please explain.

3) In vivo, the authors should also analyze mitochondrial dysfunction and the expression of p-JNK, p-c-JUN and pp38MAPK.

Minor comments:

1) Figures 3-5, representative image of the microscope should add scale bar.

2) Figure 4, I recommend supplemental Oil Red O staining of liver tissue sections.

3) In vivo and in vitro experiments, why the author chose different pro-inflammatory factors for analysis.

4) In Figure 1B-J and Figure 5C-H, the groups represented by different colors should be marked.

Round 2

Reviewer 1 Report

In the manuscript entitled “Targeting Extracellular RNA Mitigates Hepatic Lipotoxicity 2 and Liver Injury in NASH” by Tweari et al, the authors have addressed all the queries raised at my end. I think the manuscript  has been significantly improved and is acceptable for publication.

Author Response

We are thankful to the reviewer for finding our manuscript acceptable for publication.